# Assessing the Effects of 0.3% Carboxymethylcellulose Tear Substitute Treatment on Symptoms and Signs of Dry Eye Disease in Elderly Population: A Prospective Longitudinal Study

**DOI:** 10.3390/jcm12237364

**Published:** 2023-11-28

**Authors:** Antonio Ballesteros-Sánchez, José-María Sánchez-González, Giovanni Roberto Tedesco, Carlos Rocha-De-Lossada, Gianluca Murano, Antonio Spinelli, Davide Borroni

**Affiliations:** 1Department of Physics of Condensed Matter, Optics Area, University of Seville, 41004 Seville, Spain; jsanchez80@us.es; 2Department of Ophthalmology, Clínica Novovisión, 30008 Murcia, Spain; 3Studio Oculistica Tedesco, 88024 Girifaldo, Italy; j.tedesco@libero.it; 4Qvision, Ophthalmology Department, VITHAS Almeria Hospital, 04120 Almeria, Spain; carlosrochadelossada5@gmail.com; 5Ophthalmology Department, VITHAS Malaga, 29016 Malaga, Spain; 6Regional University Hospital of Malaga, Hospital Civil Square, 29009 Malaga, Spain; 7Department of Surgery, Ophthalmology Area, University of Seville, 41009 Seville, Spain; 8Sacro Cuore—iGreco Ospedali Riuniti, 87100 Cosenza, Italy; dottgianlucamurano@gmail.com; 9Biomeeting, 89123 Reggio Calabria, Italy; antspino@hotmail.it; 10Departament of Ophthalmology, Riga Stradins University, LV-1007 Riga, Latvia; info.borroni@gmail.com; 11Centro Oculistico Borroni, 21013 Gallarate, Italy; 12Eyemetagenomics Ltd., London WC2H 9JQ, UK

**Keywords:** carboxymethylcellulose (CMC), tear substitutes, artificial tears, dry eye disease (DED), meibomian gland dysfunction (MGD)

## Abstract

Background: We aimed to evaluate the effects of 0.3% carboxymethylcellulose (CMC) tear substitute treatment in dry eye disease (DED), as well as treatment compliance and adverse events (AEs). Methods: In this prospective, longitudinal study, a total of 30 eyes receiving 0.3% CMC tear substitute four times daily for DED were evaluated. Clinical endpoints included an ocular surface disease index (OSDI) questionnaire, average non-invasive tear film break-up time (A-NIBUT), lipid layer thickness (LLT), and a Schirmer test with anesthesia (ST). Treatment compliance and AEs were also assessed. All evaluations were performed at 2, 4, and 12 weeks of follow-up. Results: At the end of the follow-up, significant improvement was observed in all clinical endpoints with the following mean values: ΔOSDI questionnaire of −22.53 ± 14.68 points, ΔA-NIBUT of 4.81 ± 2.88 s, ΔLLT of 5.63 ± 6.53 nm, and ΔST of 2.8 ± 2.1 mm (*p* < 0.001 for all comparisons). Although repeated measures analysis showed that all clinical endpoints presented statistically significant differences (*p* < 0.001 for all comparisons LLT_Baseline_–LLT_2-weeks_ (*p* = 0.460) and LLT_4-weeks_–LLT_12-weeks_ (*p* = 0.071) were the only pairs of measures that reported non-statistically significant differences). In addition, treatment compliance was 94.3 ± 5.2% and transient AEs related to the use of 0.3% CMC tear substitute were reported. Conclusions: 0.3% CMC tear substitute treatment seems to achieve beneficial effects on the OSDI questionnaire, A-NIBUT, LLT, and ST. However, further studies at this concentration are needed to confirm these results.

## 1. Introduction

Dry eye disease (DED) is a pervasive and escalating global health concern [1,2], which affects approximately 10–20% of the adult population [2] and is characterized by a multitude of symptoms, such as ocular discomfort, visual disturbances, and persistent soreness that reduce patients’ quality of life [3,4,5]. The tear film is a complex, multi-layered structure essential for ocular surface health [6]. It comprises three distinct layers: the outermost lipid layer, produced by the meibomian glands, prevents tear evaporation; the middle aqueous layer, secreted by the lacrimal glands, provides moisture and nutrients; and the inner mucous layer, produced by conjunctival goblet cells, helps in tear distribution and ocular surface adherence. The multifactorial nature of DED presents a significant challenge to healthcare providers [7]. The intricacies of DED extend to tear film instability, ocular surface inflammation, and feedback mechanisms involving tear hyperosmolarity [8,9]. In addition, DED may coexist with conditions, such as meibomian gland dysfunction (MGD) or autoimmune disease, complicating management strategies [10]. These challenges suggest the need for novel therapeutic agents that target specific mechanisms involved in the pathogenesis of DED [8,9,10]. 

Carboxymethylcellulose (CMC), a chemically modified cellulose derivative, has emerged as a promising therapeutic agent [11]. As an anionic polymer composed of glucopyranose subunits [12], CMC offers unique hydrophilic, viscoelastic, and bioadhesive properties [13], which have been lauded for their potential in ophthalmological applications [14,15,16,17,18,19,20]. CMC mimics the muco-mimetic properties of natural mucin found in the tear film, thus aiding in moisture retention and corneal epithelium healing [12,21]. In addition, CMC may also reduce the enrichment of *Firmicutes* bacteria, which is associated with MGD [22,23]. Recent innovations, such as preservative-free CMC tear substitutes, have expanded the scope of its application, significantly reducing the toxic effects commonly associated with preservatives in ophthalmic solutions [19]. In addition, several randomized controlled studies (RCTs) have also reported the efficacy and safety of CMC tear substitute treatment in improving the ocular surface and stabilizing the tear film, thus providing a tangible reduction in DED symptoms and signs [14,15,16,17,18,19,20]. However, these studies used 0.5% and 1% CMC tear substitute treatment, which is associated with greater transient blurred vision [17,19], affecting treatment compliance [24].

Therefore, the aim of this study is to analyze the effects of 0.3% CMC tear substitute treatment on DED symptoms and signs, as well as treatment compliance and adverse effects (AEs) after its instillation. This original research is intended to fill existing gaps in the understanding of the role of CMC in DED treatment by conducting a robust, evidence-based evaluation.

## 2. Materials and Methods

### 2.1. Study Design and Participants

This prospective, longitudinal study was carried out at the Tedesco Eye Center (Girifalco, CZ, Italy), between November 2022 and February 2023. This study fulfilled all the requirements of the Declaration of Helsinki and was approved by the center’s internal review board (Approval Nr. 03/2022). Before initiating the study, informed consent was obtained from each patient. The inclusion and exclusion criteria are shown in Table 1.

### 2.2. Treatment and Clinical Endpoints

Thirty eyes of 30 Caucasian patients, 7 (23.3%) men and 23 (76.7%) women, with a mean age of 74.16 ± 6.58 (67–92) years, were enrolled in the study. In addition, there was no loss to follow-up in the study. Demographic characteristics are shown in Table 2. Patients were instructed to instill 1 drop of 0.3% CMC (Clarastill^®^, Bruschettini SRL, Genova, Italy) into each eye 4 times daily for 12 weeks. Carboxymethylcellulose tear substitutes are commercially available in various formulations, including solutions and gels, with varying viscosities. These formulations are designed to cater to different severity levels of dry eye disease, providing options for personalized patient care. The artificial tears used in this study comprised carboxymethylcellulose 0.3%, methyl p-hydroxybenzoate, propyl p-hydroxybenzoate, glycerin, N-acetylcarnosine, sodium chloride, edetate disodium, sodium tetraborate, potassium bicarbonate, and purified water.

The assessment of clinical endpoints was carried out in the sequence proposed by Ballesteros et al. [27] to best preserve the integrity of the tear film to avoid affecting test results: (1) OSDI questionnaire (expressed in points); (2) average NIBUT (A-NIBUT, expressed in seconds), which was measured with the Keratograph^®^ M5 (Oculus Optikgeräte GmbH, Wetzlar, Germany); (3) lipid layer thickness (LLT, expressed in nanometers), which was assessed with the Lipiview^®^ II ocular surface interferometer (Johnson & Johnson, New Brunswick, NJ, USA); and (4) Schirmer test with anesthesia (ST, expressed in millimeters).

#### 2.2.1. Ocular Surface Disease Index Questionnaire

The OSDI is a 12-item questionnaire designed to provide a rapid assessment of the symptoms of ocular irritation consistent with dry eye disease and their impact on vision-related functioning [25]. The 12 items are graded on a scale of 0 to 4, where 0 indicates none of the time; 1, indicates some of the time; 2, indicates half of the time; 3, indicates most of the time; and 4, indicates all of the time [25,28]. For items 6 to 12, the option “not answered” is also available. The total OSDI score, ranging from 0 (no ocular surface disease) to 100 (severe ocular surface disease) points, is calculated on the basis of the following formula [28]:OSDI = {[sum of the scores for all questions answered × 100]/total number of questions answered} × 4

#### 2.2.2. Average Non-Invasive Tear Film Break-Up Time

A-NIBUT is a measure of tear film stability [27]. The Keratograph^®^ M5 (Oculus, Wetzlar, Germany) evaluates NIBUT automatically by projecting Placido rings onto the corneal surface, indicating the time taken for the first disruption of the tear film after a blink [29]. Three consecutive measurements were averaged for statistical analysis.

#### 2.2.3. Lipid Layer Thickness

LLT is also a measure of tear film stability [27]. The Lipiview^®^ (Johnson & Johson Vision Care, San Francisco, CA, USA) II ocular surface interferometer evaluates LLT automatically assesses LLT with nanometer precision by recording a 20 s video of the tear film interference pattern and then displays the data in interferometric color units (ICU), where 1 ICU reflects approximately 1 nm of LLT [30,31].

#### 2.2.4. Schirmer Test

Schirmer test with anesthesia (ST) quantifies tear production [27]. The patient is instructed to look up and the test strip is placed between the palpebral conjunctiva of the lower eyelid and the bulbar conjunctiva. Subsequently, the patient is asked to keep the eyes gently closed for five minutes. After this time, the test strip can be removed and the Schirmer test score is determined by the length of the moistened area of the strip [32].

#### 2.2.5. Adverse Events

Regarding safety endpoints, ocular or systemic AEs related to the treatment were evaluated. AEs were systematically evaluated through patient-reported symptoms and clinical examinations at each follow-up visit. This approach allowed for the timely identification and documentation of any treatment-related complications. Compliance with treatment was also assessed using patient dosing diaries. Compliance was calculated as the total number of doses that should have been administered multiplied by 100 [33]. All clinical endpoints were assessed at screening, baseline (day 1), and 3 follow-up visits: (1) week 2 (15 ± 2 days), (2) week 4 (30 ± 2 days), and (3) week 12 (90 ± 2 days). In addition, they were obtained in standard environmental conditions in the same room by a trained optometrist.

### 2.3. Statistical Analysis

Statistical analyses were performed with SPSS statistics software, version 28.0 (IBM Corporation, Armonk, NY, USA). A sample size of 27 patients was estimated by the paired mean (repeated measures in one group) using the GRANMO calculator, version 7.12 (Municipal Institute of Medical Research, Barcelona, Spain). Estimation was based on a statistically significant paired difference at 95% confidence and with 80% power of 7.03 ± 2.9 s in NIBUT based on previous studies [14,15,16,17,18,19,20,22]. Continuous variables were displayed as the mean ± standard deviation (SD), while ordinal categorical variables were expressed as frequencies (*n*) and percentages (%). Before the analyses, one eye was randomly selected. The randomization scheme was generated using an online randomizer program https://www.randomization.com, accessed on 12 November 2023. After testing for normality with the Shapiro–Wilk test, the increment (Δ) was calculated to evaluate the treatment efficacy. It was defined as the change from the baseline (B) to the last visit (LV) “Δ = LV − B”. A repeated-measures ANOVA (parametric) analysis was performed. In addition, a post hoc analysis with Bonferroni adjustment was performed to determine statistically significant differences between pairs of measures. The level of significance was *p* < 0.05 for all comparisons.

## 3. Results

### Clinical Endpoints Outcomes

The effectiveness of 0.3% CMC tear substitute treatment in DED is shown in Table 3. After 12 weeks of follow-up, patients reported a significant improvement in ΔOSDI, ΔA-NIBUT, ΔLLT, and ΔST with a mean value of −22.53 ± 14.68 points, 4.81 ± 2.88 s, 5.63 ± 6.53 nm, and 2.8 ± 2.1 mm, respectively (*p* < 0.001 for all comparisons). In the repeated-measures ANOVA analysis, all clinical endpoints showed statistically significant differences (*p* < 0.001 for all comparisons). In addition, the post hoc analysis with Bonferroni adjustment revealed that LLT_Baseline_–LLT_2-weeks_ (*p* = 0.460) and LLT_4-weeks_–LLT _12-weeks_ (*p* = 0.071) reported non-statistically significant differences. The remaining pairs of measures for OSDI, A-NIBUT, LLT, and ST showed statistically significant differences (*p* < 0.001).

According to the doses recorded in the dosing diaries, treatment compliance was 94.3 ± 5.2%. In addition, no systemic AEs related to the use of 0.3% CMC tear substitute were reported during the follow-up. However, four patients (13.3%) reported ocular irritation during the first 10 min after 0.3% CMC tear substitute instillation.

## 4. Discussion

### 4.1. Carboxymethylcellulose Efficacy

In this study, ΔOSDI questionnaire, ΔA-NIBUT, ΔLLT, and ΔST improved significantly after 12 weeks of follow-up [34]. In addition, all pairs of repeated measures showed significant differences, except for LLT_Baseline_–LLT_2-weeks_ and LLT_4-weeks_–LLT_12-weeks_. However, these pairs of measures obtained *p*-values with the Bonferroni adjustment of 0.460 and 0.071, minimizing the α-type error.

Several randomized clinical trials have evaluated the effects of CMC tear substitute treatment in DED [14,15,16,17,18,19,20]. Baudouin et al. [16], Aragona et al. [19], and Salim et al. [20] reported a significant ΔOSDI questionnaire improvement of −12.22 ± 6.47 points after 10 ± 2 weeks of follow-up. However, our study achieved a significant ΔOSDI questionnaire improvement of −22.53 ± 14.68 points. This difference of −10.31 points could be attributed to the lower OSDI questionnaire baseline values reported by Baudouin et al. [16], Aragona et al. [19], and Salim et al. [20]. Regarding ΔNIBUT, Lee et al. [15] and Salim et al. [20] achieved a significant improvement of 2.19 ± 0.31 s after 8 weeks of follow-up, while our study showed a significant improvement of 4.81 ± 2.88 s. This difference of 2.62 s may be explained by variations in the duration of follow-up between the studies. A similar situation occurs with ΔST since our study obtained a value 1.07 mm higher than that reported by Salim et al. [20]. In addition, our improvements in DED symptoms and signs could be sustained until 1 year of follow-up, as reported by Bruix et al. [14]. It is important to mention that Bruix et al. [14], Lee et al. [15], Baudouin et al. [16], Aragona et al. [19], and Salim et al. [20] also showed that CMC tear substitute treatment achieved a higher reduction in DED symptoms and signs compared to sodium chloride (NaCl), sodium hyaluronate (SH), and hyaluronic acid (HA) tear substitutes [14,15,16,19,20]. Consequently, different studies have also evaluated the effects of CMC tear substitute treatment in patients with DED after cataract [17] and LASIK surgery [18], obtaining significant improvements in DED symptoms and signs at 4 and 12 weeks of follow-up, respectively. These aforementioned studies also evaluated the total corneal fluorescein staining (tCFS), achieving significant improvements at the end of the follow-up periods. However, tCFS was not assessed in our study due to the influence of the anesthetic on corneal staining [35], which was used to assess the ST.

Although the mechanism of action of CMC is not yet fully understood, it has been shown that its high micro-viscosity allows it to bind to human corneal epithelial cells [12], improving the ocular protection index for at least 20 min after instillation [21]. In addition, CMC also reduces the enrichment of *Firmicutes* bacteria [22], which have been associated with MGD and, therefore, a reduced LLT [22,23]. This is consistent with our results, which seem to be higher than those reported by the aforementioned studies that used 0.5% CMC tear substitute treatment. Therefore, this highlights the benefits of CMC at low concentrations, but further studies are needed to confirm these results.

### 4.2. Carboxymethylcellulose Safety

In this study, no systemic AEs were reported after instillation of 0.3% CMC tear substitute treatment. However, ocular irritation was reported in 13.3% of patients during the first 10 min after instillation. Similar results have been reported by Yao et al. [17] and Aragona et al. [19], with AEs in 7.9 ± 2.1% of patients who received 0.5% CMC tear substitute treatment, respectively. The most commonly reported AEs were eye irritation, blurred vision, and foreign body sensation. In addition, these AEs are similar to those reported with other artificial tears [36] and disappeared within 5 min of instillation. In addition, Lee et al. [15], Baudouin et al. [16], Wallerstein et al. [18], and Salim et al. [20] did not report AEs after instillation of 0.5% CMC tear substitute treatment. Therefore, 0.3% and 0.5% CMC tear substitute treatment seems to be safe for DED, but the possibility of mild and transitory AEs should be considered.

### 4.3. Strengths and Limitations

According to the best of our knowledge, this is the first study to analyze the effects of CMC tear substitute treatment at 0.3% in an elderly population with DED. However, there are some limitations that need to be addressed. The absence of a placebo group may influence the validity of the results, making it difficult for researchers to make strong claims about the effectiveness and safety of the 0.3% CMC eye drop treatment. In addition, despite the sample size calculation, the number of patients included may be small, leading to less accurate results. tCFS, which is usually the primary endpoint in DED studies, was not assessed in this study. However, this is due to the anesthesia used for ST assessment, which may increase corneal staining [35]. Therefore, it would be of interest for future studies to evaluate tear volume by objective and non-invasive tests, such as tear meniscus height (TMH) and area (TMA) to avoid the influence of traditional tests on tCFS [27]. Recently, it has been shown that some tear substitutes can improve meibomian gland function [37]. This study does not address this issue, thus new studies analyzing the effects of 0.3% CMC tear substitute on the meibomian glands are needed. Overall, there is a need for larger, well-designed, strictly blinded, randomized clinical trials evaluating the long-term effects of 0.3% CMC tear substitute treatment in patients with DED. In addition, it would also be interesting to compare the effects of CMC tear substitute treatment at different concentrations, as well as their combination with other compounds such as HA or cross-linked HA. This would be of special interest in patients with Sjogren’s syndrome and MGD, which are the main causes of aqueous-deficient and evaporative dry eye, respectively. While this study focused on the assessment of the lipid layer thickness, the muco-aqueous layer thickness was not measured. This decision was based on the current study’s emphasis on the lipid layer’s role in dry eye disease and the logistical constraints of implementing additional imaging techniques like the Tear Film Imager (TFI) in our study setting.

## 5. Conclusions

In conclusion, 0.3% CMC tear substitute treatment seems to achieve beneficial effects on DED symptoms and signs in the elderly population. This treatment significantly improves the OSDI questionnaire, A-NIBUT, LLT, and ST with high compliance and transient AEs after instillation. Therefore, further studies at this concentration are warranted to validate our findings.

## Figures and Tables

**Table 1 jcm-12-07364-t001:** Inclusion and exclusion criteria.

Inclusion Criteria
(1) Age greater than 65 years old (2) DED diagnosis according to DEWS II, meeting at least one of the following conditions [25]: OSDI score greater than 13 points. NIBUT less than 10 s. (3) MGD diagnosis according to the International Workshop on MGD, meeting at least one of the following conditions [26]: Irregularity of the eyelid margin or mucocutaneous junction. Vascularity of the eyelid margin. Plugged or capped Meibomian gland orifices. Meibomian gland atrophy. Decreased meibum quality and quantity.
**Exclusion criteria**
(1) Patients with degenerative diseases that make topical application of treatment difficult, such as Parkinson’s disease or multiple sclerosis. (2) All corneal disorders that prevent diagnostic tests from being performed, including: Active corneal infections. Corneal dystrophies. (3) Active ocular allergy. (4) Contact lens wearers. (5) Pregnant or lactating women. (6) Patients who did not understand or comprehend informed consent.

DED, Dry eye disease; DEWS II, Drye eye workshop II; MGD, Meibomian gland dysfunction; NIBUT, Non-invasive tear film break-up time; OSID, Ocular surface disease index.

**Table 2 jcm-12-07364-t002:** Demographics characteristics.

Mean ± SD (IQR) or *n* (%)	*n* = 30
Age (years)	74.16 ± 6.58 (67–92)
Sex, male/female	7 (23.3)/23 (76.7)
Race, Caucasian	30 (100)

IQR, interquartile range; SD, standard deviation.

**Table 3 jcm-12-07364-t003:** Changes in clinical endpoints during follow-up visits.

Variables ^1^	Follow-Up				
	Baseline	2 Weeks	4 Weeks	12 Weeks	*p*-Value
OSDI, points	36.78 ± 19.16	28.67 ± 13.67	21.2 ± 9.3	14.23 ± 5.91	<0.001 *^, 2^
	(2–77)	(12–63)	(11–44)	(7–32)	
A-NIBUT, s	6.23 ± 2.44	7.75 ± 3.56	9.36 ± 4.1	11.04 ± 4.48	<0.001 *^, 2^
	(2–9.7)	(2–15)	(3.4–18)	(5.5–20)	
LLT, nm	78.8 ± 16.12	80 ± 14.75	82.87 ± 12.66	84.33 ± 11.31	<0.001 *^, 2^
	(52–100)	(55–100)	(60–100)	(65–100)	
ST, mm	8.73 ± 1.99	9.46 ± 2.44	10.2 ± 2.31	11.53 ± 3.01	<0.001 *^, 2^
	(6–15)	(6–15)	(7–15)	(7–15)	

LLT, lipid layer thickness; mm, millimeter; nm, nanometer; NIBUT, non-invasive tear film break-up time; OSDI, ocular surface disease index; s, seconds; ST, Schirmer test. ^1^ Expressed as mean ± standard deviation (SD) with interquartile ranges (IQRs); ^2^ Repeated measures ANOVA with Bonferroni adjustment. * Statistical significance level of <0.05.

## Data Availability

The data presented in this study are available on request from the corresponding author. The data are not publicly available due to their containing information that could compromise the privacy of research participants.

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
