# Peer review of "Assessing the Effects of 0.3% Carboxymethylcellulose Tear Substitute Treatment on Symptoms and Signs of Dry Eye Disease in Elderly Population: A Prospective Longitudinal Study"

_jcm, 2023, doi:10.3390/jcm12237364_

Round 1
Reviewer 1 Report
Comments and Suggestions for Authors
The authors presented an interesting longitudinal study evaluating the effects of 0.3% Carboxymethylcellulose Tear Substitute Treatment in Dry Eye Disease patients. The manuscript is clear and the conclusions are consistent with the evidence presented. The manuscript is with merit and the findings are worth reporting, but the authors should address the following comments:
- Introduction: it would be useful to have a brief anatomical description of the tear film with its layers for the readers that are not familiar with it, as the journal is Journal of Clinical Medicine
- Since “Meibomian Gland Dysfuncition (MGD)" has been mentioned at line 56, only the abbreviation “MGD” should be used at line 66
- Line 88: the explanation for the abbreviation “DEWS II” should be indicted
- Lines 103-109: it would be useful that the authors expand slightly the section on clinical endpoints by providing a brief description of what the ocular surface disease index (OSDI) questionnaire, non-invasive tear film break-up time (A-NIBUT), and Schirmer test with anesthesia (ST) are, for readers that are not ophthalmologists and are not familiar with them, as the journal is Journal of Clinical Medicine
- How were the adverse events evaluated? The authors should add this information to the corresponding sentence at line 110
- The authors measured the lipid layer thickness, but not the muco-aqueous layer thickness (that can be assessed for example with imaging like the Tear Film Imager (TFI), AdOM): was there a rationale behind the fact that it was not measured? The authors should discuss this point in the limitations section.
- What are the forms of Carboxymethylcellulose Tear Substitute currently commercially available? The authors should mention this information.
Reviewer 2 Report
Comments and Suggestions for Authors
The authors evaluated the effectiveness of a low concentration of CMC (0.3%) for the first time, which, although not groundbreaking from a scientific standpoint, I consider important from a clinical perspective to generate evidence for all available treatments for dry eyes. Overall, I find the study very interesting but with many limitations, some of which cannot be addressed (such as the lack of a control group), and others that I suggest in the major concerns that the authors must to address. Despite the limitations in the study design, I believe the work could be suitable for publication as long as the results related to ocular surface damage, tear meniscus (since measurements were taken with Keratograph), and Meibomian glands (since measurements were taken with LipiView) are included. The inclusion of these results is crucial for accepting the manuscript for publication after this initial review, so if the authors do not include these results, I consider the work lacking in terms of its scientific methodology.
INTRODUCTION
· Lines 72-74: Please do not consider transient blurred vision (around 1-3 minutes after instillation) as an adverse effect of high-viscosity CMC.
METHODS
· Lines 87-90: Achieving at least one of the two diagnostic criteria is not a criterion of the TFOS DEWS II diagnosis report. Please clarify and consider rewriting the paragraph on inclusion and exclusion criteria (perhaps a table would be more effective) as it is a bit cumbersome.
· Lines 101-102: Please provide the composition of the artificial tears.
· Lines 103-113: You need to provide much more detailed information about each procedure separately.
· MAJOR CONCERN: Since you have the Keratograph available to perform the measurements of NIBUT, why didn't you report the results of the rest of the dry eye-related parameters incorporated in this topographer? This includes the measurement of tear meniscus height, the current gold standard for assessing tear volume (instead of the Schirmer test). The same applies to LipiView and all parameters related to MG. You MUST include this data in the manuscript.
· Lines 120-122: Please provide more information about the sample size calculation. Which study are you referring to? What test did you select with the GRANMO calculator?
RESULTS
· Lines 136-138: While it's common in ophthalmology to incorporate demographic data in Results, I suggest including them in Methods before or after inclusion and exclusion criteria.
· Since you provide the standard deviation, the inclusion of interquartile ranges doesn't provide any extra information, so please remove these values to improve the clarity of the manuscript.
· MAJOR CONCERN: The authors are duplicating results in Figure 1 and Table 2, but at the same time, they are providing a different statistical analysis (not specified in Figure 1), which is not correct. Please remove Figure 1. If you have a follow-up of different visits (baseline, 2, 4, and 12 weeks), you need to perform ANOVA for related samples and, once you verify the statistical significance, perform the Bonferroni correction to provide pairwise comparison between visits (not only the comparison between the baseline and the final visit). Repeated measures ANOVA with Friedman’s test is not an existing test in statistics. Please reanalyze this data and provide both the ANOVA significance of the four visits and the Bonferroni correction (pairwise comparison between visits).
· Line 151: P = 0.058.
· Lines 152-153: What AEs did you evaluate? How did you evaluate them? Please provide more information in Methods and Results.
DISCUSSION
· Lines 156-161: Please delete. This is not part of the discussion of your results and was previously stated in the Introduction.
· Lines 165-166: I don’t consider the sample small at all, so this non-significant value is not relevant to your results. What is important is to provide the Bonferroni correction p-values to confirm that you are minimizing type 1 errors since you are doing multiple comparisons.
· Line 203: Change RCTs to randomized clinical trials.
· Lines 203-224: The numbered results of different authors are not interesting for the manuscript. What is interesting is a critical discussion of your results in comparison with the main findings of these authors. Summarize the main findings of these studies instead of reporting their results separately (I had difficulties reading this paragraph).
· MAJOR CONCERN (Lines 225-232): Those ideas are not consistent with your results since you didn’t provide results about ocular surface damage. I didn’t realize until now, but the results of ocular surface damage (at least staining) MUST be provided in the manuscript.
· Lines 234-243: See the comments about Lines 203-224.
CONCLUSIONS
· Lines 266-267: "with high compliance and no AEs after instillation" must be supported with quantitative data included in the Results. Otherwise, remove it.
Comments on the Quality of English Language
The clarity of the manuscript (language and format) could be improved.
Author Response
Dear editor and reviewer, the authors are grateful for the extensive review of the manuscript. Authors also suggest that you reconsider the manuscript again for publication, as several changes have been incorporated in the following sections:
- Methods: Creating a new table for inclusion and exclusion criteria.
- Results: Re-analyzing the data with repeated measures ANOVA and bonferroni adjustment.
- Discussion: Summarizing the results of different studies by performing a critical analysis in comparison with our results.
For a better understanding of the changes performed, Please see the attachment.

Round 2
Reviewer 2 Report
Comments and Suggestions for Authors
The authors have significantly improved the initial version of the manuscript. It could be accepted in its current version.